# Review of Regional Therapies for Gastric Cancer with Peritoneal Metastases

**DOI:** 10.3390/cancers14030570

**Published:** 2022-01-23

**Authors:** Beatrice J. Sun, Byrne Lee

**Affiliations:** Department of Surgery, Stanford University, Stanford, CA 94305, USA; sunbj@stanford.edu

**Keywords:** gastric cancer, peritoneal metastases, intraperitoneal chemotherapy, HIPEC, PIPAC, cytoreductive surgery, peritoneal carcinomatosis index

## Abstract

**Simple Summary:**

Gastric cancer is usually diagnosed at late stages and is associated with poor five-year survival rates. Metastasis to the peritoneal cavity is common and leads to even worse outcomes. Currently, the mainstay of treatment for metastatic gastric cancer is systemic chemotherapy or supportive care. These recommendations remain despite evidence that suggests systemic therapy has poor penetration into the abdominal cavity, limiting efficacy against peritoneal disease. Newer treatments have been developed to address this problem, specifically regional therapies aimed at delivering chemotherapy directly into the peritoneal cavity to eradicate tumor cells. These novel therapies include hyperthermic intraperitoneal chemotherapy, normothermic intraperitoneal chemotherapy, and pressurized intraperitoneal aerosolized chemotherapy. Regional therapies may also be combined with surgery to remove both macroscopic and microscopic disease. Although more clinical trials are needed to evaluate its efficacy, early studies have shown promising outcomes with intraperitoneal chemotherapy.

**Abstract:**

Gastric cancer carries a poor prognosis and is a leading cause of cancer-related mortality worldwide. Patients with gastric cancer who develop peritoneal metastases have an even more dismal prognosis, with median survival time measured in months. Since studies have demonstrated that systemic chemotherapy has poor penetration into the peritoneum, multimodal treatment with intraperitoneal chemotherapy has been proposed for the treatment of peritoneal metastases and has become the foundation for newer therapeutic techniques and clinical trials. These include heated intraperitoneal chemotherapy (HIPEC) with cytoreductive surgery (CRS), which involves the application of heated chemotherapy into the abdomen with or without tumor debulking surgery; normothermic intraperitoneal chemotherapy (NIPEC), in which non-heated chemotherapy can be delivered into the abdomen via a peritoneal port allowing for repeat dosing; and pressurized intraperitoneal aerosolized chemotherapy (PIPAC), a newer technique of pressurized and aerosolized chemotherapy delivered into the abdomen during laparoscopy. Early results with intraperitoneal chemotherapy have shown promise in increasing disease-free and overall survival in select patients. Additionally, there may be a palliative effect of these regional therapies. In this review, we explore and summarize these different intraperitoneal chemotherapy treatment regimens for gastric cancer with peritoneal metastases.

## 1. Introduction

Gastric cancer is one of the leading causes of cancer deaths, with over 750,000 deaths annually [1]. The impact of gastric cancer includes low survival, late stage at diagnosis, and frequent metastasis. Peritoneal metastases occur in 30% of advanced gastric cancer diagnoses and carries a dismal prognosis [2,3,4]. Current National Comprehensive Cancer Network (NCCN) guidelines for advanced gastric cancer include palliative management with chemoradiation, systemic therapy, or supportive care [5]. Unfortunately, standard therapies such as systemic chemotherapy and immunotherapy have limited efficacy in gastric cancer with peritoneal metastases. Therefore, there is no effective treatment outside clinical trials.

Surgical management of peritoneal carcinomatosis from gastric cancer has been a controversial topic. The REGATTA trial aimed to determine if gastrectomy would be beneficial compared to systemic chemotherapy in the setting of limited metastatic disease. The study demonstrated no survival benefit of surgical resection of the primary tumor when compared to palliative chemotherapy alone [6]. However, it did not attempt to address survival in the setting of complete resection of all metastatic disease.

Prior studies have demonstrated poor penetration and response of systemic chemotherapy in peritoneal disease, with overall survival of 5–11 months, often with significant systemic toxicity [7,8,9,10,11]. Its limited efficacy in the peritoneum, especially in the setting of gastric cancer, led to further interest and investigation into intraperitoneal chemotherapy. Advantages of this form of regional therapy include its direct application into the peritoneal cavity, which allows for the exposure of greater drug concentration to regional tumor cells for a longer period of time while maintaining a low systemic concentration, and thus carries a low risk for systemic toxicity [12,13,14]. The peritoneal-plasma barrier results in a slower peritoneal drug clearance compared to plasma drug clearance [13]. Despite the advantage of low systemic toxicity and decreased drug clearance, passive penetration of intraperitoneal chemotherapy is only 1–3 mm [13,14,15]. Thus, to improve response, intraperitoneal therapies should ideally be combined with the elimination of macroscopic disease via surgical resection or by repeat intraperitoneal chemotherapy dosing locally [12,14].

In this review, we summarize three different types of intraperitoneal chemotherapy: hyperthermic intraperitoneal chemotherapy (HIPEC) with cytoreductive surgery (CRS), which involves the application of heated chemotherapy into the abdomen with or without tumor debulking surgery; normothermic intraperitoneal chemotherapy (NIPEC), in which non-heated chemotherapy can be delivered into the abdomen via a peritoneal port and allows for repeat dosing; and pressurized intraperitoneal aerosolized chemotherapy (PIPAC), a newer technique of pressurized and aerosolized chemotherapy delivered into the abdomen during laparoscopy, in the treatment and palliation of gastric cancer.

## 2. HIPEC and CRS

### 2.1. History

HIPEC and CRS have been described in earlier studies as treatments for peritoneal surface malignancies. One of the earliest case reports was from Spratt et al. in 1980, in which HIPEC was used to treat a patient with pseudomyxoma peritonei [16]. The authors had previously developed a thermal infusion filtration unit as a delivery system to treat malignant ascites and peritoneal metastases [17]. In this patient, following cytoreductive surgery, the thermal infusion filtration system was used to deliver heated lactated ringers, followed by hyperthermic chemotherapy, into the peritoneal cavity. The patient subsequently recovered without complication and remained active and asymptomatic at 8-month follow-up. In addition, early studies in Japan in the late 1980s presented patients with advanced gastric cancer who underwent cytoreductive surgery followed by HIPEC with mitomycin C via peritoneal catheters, demonstrating the feasibility and safety of intraperitoneal chemotherapy infusion [18,19]. An example of a typical HIPEC setup is depicted in Figure 1. 

As interest and experience with HIPEC have grown, the focus has shifted to using HIPEC as a therapeutic tool to treat and reduce malignant peritoneal disease. CRS with HIPEC has shown efficacy in improving survival in appendiceal cancer, ovarian cancer, and peritoneal mesothelioma [20,21,22,23]. However, the use of HIPEC in gastric cancer with peritoneal carcinomatosis remains controversial. Despite various cohort studies that have described improved outcomes with HIPEC, there are few randomized controlled trials to date that have investigated this.

### 2.2. Randomized Controlled Trials

In 2011, Yang et al. published a phase III randomized controlled trial that evaluated the survival outcomes of patients with gastric cancer and peritoneal carcinomatosis who underwent CRS only, compared to those who underwent CRS followed by HIPEC [24] (Table 1). The study was designed to detect a 30% difference in disease-free survival and required a minimum of 60 patients to achieve adequate power. Thirty-four patients were randomized into each group: CRS alone or CRS + HIPEC. All patients underwent maximal CRS with the goal to remove all visible disease. In the CRS + HIPEC group, an open HIPEC technique was performed with cisplatin and mitomycin C at 43 °C for 60–90 min. Primary outcome of disease-free survival was 6.5 months for CRS only, compared to 11.0 months for CRS + HIPEC (*p* = 0.046). Contrary to findings in other studies [25,26], the Yang study found no difference in survival between groups in the low PCI group [24,25,26]. Interestingly, “low PCI” was defined as PCI < 20, which is a higher threshold compared to other studies and may account for the discrepancy [4,25,26]. Serious adverse events were similar between groups (11.7% in CRS, 14.7% for CRS + HIPEC) [24,27,28]. On multivariate analysis, CRS + HIPEC was associated with improved survival (HR 2.6, *p* = 0.002), highlighting the potential synergistic effect of CRS with regional therapy.

In another phase III randomized study (GYMSSA Trial), Rudloff et al. sought to investigate whether adding CRS and HIPEC to systemic chemotherapy would improve survival in patients with metastatic gastric cancer [29]. Of note, this trial included patients with non-peritoneal disease such as liver and lung metastases, which has been shown to respond better to systemic chemotherapy than peritoneal disease [35]. The intention of this study was to enroll at least 136 patients, 68 in each arm, in order to adequately power the study analysis; however, this study did not accrue the target number of patients. Of the 34 patients screened, only 16 were included: 7 underwent systemic chemotherapy alone (SA) with FOLFOXIRI, and 9 underwent CRS + HIPEC (GYMS) with a closed HIPEC technique using intraperitoneal oxaliplatin at 41 °C for 30 min. Median survival in the GYMS group was 11.3 months, compared to 4.3 months in the SA group, and while no subject in the SA arm survived over 11 months from time of randomization, 4 out of 9 GYMS subjects survived over 1 year, 2 survived over 2 years, and 1 was still alive at 4 years. All patients who survived over 1 year had PCI ≤ 15 and achieved complete cytoreduction (CCR 0) at the time of surgery. Postoperative complications for this small cohort were much higher than previously reported: 88% serious complication rate, 44% reoperation rate, and 11% 90-day mortality. Although this study was severely underpowered and survival conclusions cannot be drawn, it recognizes the potential for multimodal therapy with CRS + HIPEC to prolong survival in patients with metastatic gastric cancer, which is consistent with findings from other non-randomized studies [4,24,25,26,36].

### 2.3. Non-Randomized Studies

Glehen et al. published a multi-institutional retrospective study of patients who underwent intraperitoneal chemotherapy at 15 French institutions over the course of nearly 2 decades, from 1989 to 2007 [25]. A total of 159 patients with gastric cancer and peritoneal carcinomatosis underwent cytoreductive surgery followed by HIPEC in 94% and/or early postoperative chemotherapy in 7.5%, with either mitomycin or oxaliplatin-based regimens. Postoperative complication rate was 28%, with 14% requiring reoperation and 6.5% mortality. Improved survival was associated with high-volume centers, completeness of cytoreduction, and lower PCI score. Patients who underwent complete cytoreduction had a median survival of 15 months, with 61% surviving over 1 year and 23% over 5 years. Notably, because patients with high PCI were found to have poor long-term survival despite complete cytoreduction, the authors recommended against multimodal therapy with CRS + HIPEC in patients with PCI > 12.

A more recent multicenter study by Rau et al. evaluated data from the national German HIPEC registry between 2011 and 2016 [26]. During this time, 235 patients underwent CRS and HIPEC for gastric cancer with peritoneal metastases. Postoperative complication rate was 17%, and mortality was 5%. Their results highlighted that experienced centers that treated more than 20 patients achieved greater median overall survival compared to less experienced centers (16 months vs. 12 months, *p* = 0.02). Patients with lower PCI scores also had significantly better median overall survival after HIPEC: 18 months for PCI < 6 vs. 5 months for PCI > 15 (*p* = 0.002). 

CYTO-CHIP was another retrospective multicenter study across several French centers between 1989 and 2014 which assessed the outcomes of adding HIPEC to CRS [4]. The study included patients with either peritoneal metastases or ovarian metastases from gastric cancer primary. Of 277 patients, 180 underwent CRS + HIPEC while 97 underwent CRS only; those undergoing CRS alone tended to be older with lower PCI scores. After propensity adjustment, CRS + HIPEC was associated with improved median overall survival of 19 months vs. 12 months, and 5-year survival of 20% vs. 6%. Similar results held for adjusted recurrence-free survival, with a survival advantage in CRS + HIPEC of 14 months vs. 8 months. Overall morbidity for this study was relatively high, with a 54% rate of total complications, including anastomotic leak in 21% and 90-day mortality of 8%. A limitation of the CYTO-CHIP study was the low number of patients who met inclusion criteria. On average, less than one patient per institution per year was included. Furthermore, changing treatment paradigms during the 25-year study period make it difficult to directly compare outcomes. Additionally, there was considerable variation in treatment, including the HIPEC technique, chemotherapy regimen, and time of indwelling intraperitoneal chemotherapy. 

Two recently published phase II trials by Badgwell et al. introduced the use of laparoscopic HIPEC as a safe, well-tolerated therapeutic for patients with gastric cancer and peritoneal metastases [37,38]. Laparoscopic HIPEC can be repeated and serves as a potential preoperative tool to decrease tumor burden, demonstrate non-progression of disease, and may be used as a bridge to a complete open CRS with HIPEC. Of the 20 patients enrolled to undergo systemic chemotherapy and laparoscopic HIPEC prior to open CRS and HIPEC, there was a 35% major complication rate, 30% anastomotic leak, 50% readmission, but no 90-day mortality; overall survival was 22 months from time of diagnosis [38].

### 2.4. Overview of HIPEC Complications

Literature on HIPEC in gastric cancer with peritoneal metastases remains limited. Currently, only two randomized controlled trials have been published on this topic, one of which was terminated early due to inadequate patient enrollment [24,29]. Although several non-randomized studies demonstrate a benefit of HIPEC, the results of these studies should be interpreted with caution as they are often limited by selection bias and variable complication rates. There is still controversy regarding the widespread adoption of this technique, and thus HIPEC is still largely performed only at specialized centers.

One of the hesitations in adopting HIPEC into practice is fear of the morbidity associated with CRS + HIPEC. Reported complications include anastomotic leak, bowel perforation, bowel obstruction, fistula formation, bleeding, respiratory complications, among others [39,40]. While these complications are not unique to CRS + HIPEC, the risk tends to be higher than in colorectal or gastric surgery alone, which is expected given the aggressive nature of the procedure [41,42,43]. Complications, such as renal failure, may also be attributed to transient systemic toxicity of intraperitoneal chemotherapy. Cisplatin is commonly used during perfusion in HIPEC and is well-known to carry a 20–30% risk of acute kidney injury, sometimes with long-term sequelae. In efforts to reduce this, recent studies have examined the role of adding sodium thiosulfate and shown a significant reduction in renal injury down to 0–6% [44,45]. Overall morbidity and mortality rates have been shown to be similar between CRS only and CRS + HIPEC, and at acceptable rates, especially when performed at high-volume centers [46].

### 2.5. Survival Data in HIPEC 

The results of several studies show a survival benefit of CRS and HIPEC in patients with metastatic gastric cancer with a PCI < 12 who are able to achieve complete cytoreduction [24,25,26]. When the degree of cytoreduction is not complete, there was little to no HIPEC benefit due to high rates of recurrence. Furthermore, even in the context of complete cytoreduction, patients with high preoperative PCI > 15 demonstrate a low survival, suggesting that these patients do not benefit from CRS + HIPEC [25,26]. Because of this, laparoscopy has become an important preoperative diagnostic tool to assess PCI, a measure of the extent of peritoneal carcinomatosis. 

Laparoscopic HIPEC may be helpful in identifying patients who would benefit from cytoreductive surgery. Patients who ultimately undergo gastrectomy and debulking after laparoscopic HIPEC may result in improved survival rates [37,38]. Moreover, a patient’s baseline functional status is also a crucial preoperative consideration. Multicenter studies have demonstrated improved overall outcomes at high-volume tertiary centers, and given the complexity of careful patient selection, multidisciplinary approach, technical experience, and resources required for HIPEC, this may be prudent [25,26].

### 2.6. Chemotherapy in HIPEC

Although HIPEC techniques and regimens may vary by institution, some key principles are preserved. Studies have shown a nonuniform distribution of intraperitoneal chemotherapy within 24 h after surgery, likely due to early adhesion formation. The benefit of HIPEC as an intraoperative treatment during the time of cytoreduction mitigates the adhesion issue and increases its effectiveness as a regional therapy [47].

Hyperthermia exerts direct toxicity on cancer cells by impairing DNA repair, increasing protein denaturation, and increasing cell apoptosis. It may also act synergistically with chemotherapy, increasing the penetration depth of the drug into the tumor [12,47,48,49]. For instance, platins exhibit synergistic effects with temperatures of 41–44 °C [13,47]. The type of chemotherapy used in HIPEC has not yet been standardized, and even today, regimens are widely variable. Common HIPEC regimens include mitomycin C monotherapy [50], mitomycin C with cisplatin [24,38], cisplatin and 5FU [51], and oxaliplatin [29]. The variation of HIPEC regimens has made it difficult to validate the efficacy of treatment. This is further confounded by surgical and oncologic practices that vary widely throughout the world. Future studies are needed to identify optimal HIPEC agents and methods of delivery. 

### 2.7. Prophylactic HIPEC

Another potential application for HIPEC is in patients with advanced gastric cancer in the absence of peritoneal metastases, including those with serosal invasion and positive peritoneal cytology. Since they are at high risk for developing peritoneal carcinomatosis, HIPEC has been described for prophylactic management in this population. Desiderio et al. examined this subgroup in a meta-analysis, which showed mixed results in overall survival with or without HIPEC. The study did demonstrate lower disease recurrence in the HIPEC group (RR 0.73, *p* = 0.002), specifically in preventing peritoneal metastases (RR 0.63, *p* < 0.01) [46]. This was also associated with higher overall complications (RR 2.2, *p* > 0.01) [46]. Subsequent meta-analyses by Sun, who examined patients with serosal invasion, and Coccolini, who analyzed patients with positive cytology, corroborated the finding of decreased peritoneal dissemination after HIPEC and also noted favorable survival in the HIPEC group [36,52].

### 2.8. HIPEC in Palliation

HIPEC is also described for use in symptomatic palliation for malignant ascites. In a report in the 1980s, Fujimoto et al. had noticed that ascites regressed after HIPEC, a favorable side-effect of intraperitoneal chemotherapy [19]. Today, multiple study series have described laparoscopic HIPEC procedures for the palliation of malignant ascites from gastric cancer with peritoneal carcinomatosis. These series demonstrate an average hospital stay of 2 days and clinical resolution of ascites at 2–4 weeks in almost all patients, with no major complications or perioperative mortality reported [53,54]. One patient developed recurrent symptomatic ascites after the first procedure and underwent a second laparoscopic HIPEC with complete resolution thereafter [54]. A review by Facchiano et al. revealed that nearly 50% of patients with gastric cancer will develop malignant ascites and that 95% of them are able to achieve complete regression of debilitating malignant ascites after just a single laparoscopic HIPEC procedure [55]. This minimally invasive HIPEC technique has been adopted as a safe and effective approach for palliative management of symptomatic ascites at the end of life. 

## 3. NIPEC

### 3.1. Systemic and Intraperitoneal Chemotherapy (SIPC)

In addition to heated chemotherapy, two non-heated chemotherapy delivery systems have also been described. One such modality is known as SIPC, or bi-directional therapy, in which patients receive intraperitoneal chemotherapy infusions at regular intervals via an implantable peritoneal port, concurrently with systemic chemotherapy. This method offers repeat dosing of intraperitoneal chemotherapy, is feasible by a minimally invasive approach, and can be performed in an outpatient setting [56].

Studies evaluating NIPEC in gastric cancer with peritoneal metastases have been predominantly conducted in Asia, particularly in Japan. A number of phase II trials of SIPC using intraperitoneal paclitaxel in conjunction with various systemic chemotherapy regimens have demonstrated encouraging survival trends [31,32,57,58]. With a median of 7 to 16 weekly intraperitoneal paclitaxel infusions, these studies showed a 1-year survival rate of around 80% and a median survival time of 18 to 25 months. Peritoneal cytology converted from positive to negative in 86–97% of patients, and malignant ascites improved or resolved in over 60% [31,57,58]. SIPC was tolerated with minimal complications, mostly attributed to neutropenia.

The only randomized clinical trial on NIPEC was the PHOENIX-GC trial published by Ishigami et al. in 2018, which was highly anticipated but unfortunately did not confirm the survival advantage shown in earlier studies [30]. From 2011 to 2013, 183 patients were enrolled and randomized into the intraperitoneal and systemic chemotherapy group (IP) or the systemic chemotherapy only group (SP) in a 2:1 ratio in favor of the IP cohort. Ultimately, 114 patients were included in IP and 50 in SP. Although the study was designed to show a survival difference, the analysis failed to show a statistically significant survival advantage, with a median survival of 17.7 months in the IP group compared to 15.2 months in the SP group (*p* = 0.08). A major limitation of the study, however, was the markedly higher rate of malignant ascites in the IP cohort, suggesting potentially more advanced cancer and peritoneal disease in the IP group, although PCI was not consistently available for the study. The imbalance of the groups likely underestimates the true efficacy of the IP regimen. The 3-year survival was 21.9% in the IP group vs. 6.0% in the SP group, favoring IP group survival. Both treatments were well-tolerated with acceptable complications rates due to hematologic side effects.

Conversion gastrectomy was discussed in phase II trials as a potential adjunct for select patients if SIPC was successful in decreasing their peritoneal disease burden. In fact, as many as 40–60% of study patients undergoing SIPC eventually underwent surgical resection [31,32,57,58]. Indications to proceed with gastrectomy included: negative peritoneal cytology, resolution or regression of visible peritoneal metastases on repeat diagnostic laparoscopy, and no extraperitoneal metastases [59,60,61]. Several studies have shown positive results with this approach. A median of 5–8 courses of neoadjuvant therapy (intraperitoneal paclitaxel and systemic chemotherapy) were administered, after which patients with good functional status who met criteria underwent gastrectomy and lymphadenectomy. HIPEC was not performed. R0 resection was achieved in 65–70% of patients, with 1-year survival as high as 100% in some reports [58,59,60,61]. Median survival time for the patients undergoing gastrectomy was on the order of 26 to 30 months, compared to 12–13 months in the non-surgery group. A few studies describe continuing bi-directional therapy after gastrectomy, although the length of treatment and long-term outcomes have not been examined [59,61]. Not surprisingly, patient selection plays a key role in treatment modality, as not all patients are candidates to undergo gastrectomy in this context. No studies to date have directly compared the use of CRS + HIPEC with SIPC + conversion gastrectomy, but preliminary cohort studies have shown the latter as a potentially promising treatment option.

### 3.2. Early Postoperative Intraperitoneal Chemotherapy (EPIC)

Another application of non-heated intraperitoneal chemotherapy is EPIC, a catheter-based chemotherapy utilized in the adjuvant setting following CRS with or without gastrectomy. The rationale is that repeat dosing of intraperitoneal chemotherapy in the early postoperative period, prior to significant adhesion formation, may prevent early recurrence of disease in advanced gastric cancer at risk for peritoneal metastases. 

Reported EPIC protocols are heterogeneous, and while several studies report a survival advantage with the addition of adjuvant EPIC, overall outcomes have been mixed [62]. A randomized trial by Yu et al. published in 2001 examined the efficacy of gastrectomy with adjuvant EPIC compared to gastrectomy alone in patients with advanced gastric cancer [63]. EPIC protocol in this study consisted of a 23-h infusion of mitomycin C on postoperative day 1, followed by daily infusions of fluorouracil (5-FU) on postoperative days 2–5. The EPIC group showed an improved overall survival rate at 5 years, 54% vs. 38%, particularly in patients with more advanced stage gastric cancers; however, complications such as intra-abdominal bleeding and abscess were significantly more pronounced in the study group. Cheong et al. described another EPIC protocol after gastrectomy, consisting of 5-FU and cisplatin for 4 days at a time, repeated on a monthly basis [33]. The authors found that patients who underwent R0 resection were able to receive more cycles of EPIC and had better median survival, but it was not possible to determine whether the survival benefit was due to complete cytoreduction or the EPIC treatment. In 2018, a phase II trial, known as INPACT, was performed to compare postoperative intraperitoneal chemotherapy versus intravenous chemotherapy [34]. This trial failed to show any advantage in survival or recurrence rates in the EPIC arm. Thus, the role of EPIC as an adjuvant treatment remains controversial.

### 3.3. Chemotherapy in NIPEC

Chemotherapy regimens in NIPEC, unlike in HIPEC, are relatively consistent. Initial studies used cisplatin and mitomycin C based on prior HIPEC experience, but further pharmacokinetic studies demonstrated that these drugs exhibit a low peritoneal to plasma ratio, indicating that these chemo drugs have a high systemic and low peritoneal absorption [64]. Instead, this shifted to favor the use of taxanes including paclitaxel and docetaxel for intraperitoneal chemotherapy, which have high molecular weight and lipophilic structures allowing for slow absorption and, therefore, long intraperitoneal retention time without the need for heat augmentation [12,13]. First shown to be effective against ovarian cancer, today, taxanes are the chemotherapy drugs of choice in NIPEC protocols [65].

The ideal window for postoperative NIPEC is with peritoneal lavage in the first 5 days postoperatively, before fibrosis develops and walls off the cancer cells in the abdomen [66]. NIPEC is always given concurrently with systemic chemotherapy, in order to target the peritoneum via both subperitoneal capillaries and passive diffusion [48]. The intraperitoneal chemotherapy itself has poor penetration into tissue, and paclitaxel penetrates only 100–200 µm deep; thus, repeated intraperitoneal dosing is necessary to penetrate tumor tissue and exert antitumor effects [56]. Taxanes have the added benefit of not producing adhesions due to their antiproliferative effect, which allows for repeat dosing and even distribution, and patients do not experience chemical peritonitis [67]. Typical NIPEC protocols include an oral-systemic therapy for 2 weeks (treatment days 1–14), with paclitaxel given both intraperitoneally and intravenously on days 1 and 8. The efficacy and safety of intraperitoneal taxanes have been shown in many phase II trials in Asia, with positive results and increased survival [31,32,57].

Unlike CRS + HIPEC, which is usually only performed once, NIPEC can be dosed repeatedly for increased intraperitoneal penetration and efficacy. Overall, the procedure is well-tolerated, with low complication rates, and offers reasonably increased survival in most cases compared to systemic therapy.

## 4. PIPAC

### 4.1. Method and Use

Pressurized intraperitoneal aerosol chemotherapy (PIPAC) is a newer modality of delivering intraperitoneal chemotherapy that has garnered attention over the last decade. The idea of pressurized chemotherapy was introduced in 2012 when a German group published on a novel device that could inject pressurized, aerosolized material into the peritoneal cavity of pigs, demonstrating improved distribution and penetration compared to a peritoneal lavage control [68]. The use of the device, CapnoPen (CapnoMed, Zimmern ob Rottweil, Germany), has provided a novel way of introducing intraperitoneal chemotherapy for the treatment of peritoneal carcinomatosis [69].

Several advantages of PIPAC as an alternative therapeutic modality have been proposed, including more equal distribution within the peritoneum, enhanced uptake and deeper penetration into the tumor, and minimally invasive access allowing for repeat interval dosing [68,69,70]. In addition, because of the pressurized administration, only 1/10 of the chemotherapy dose of HIPEC is needed, which further prevents the risk of toxicity with repetitive doses [69]. In patients who are able to undergo multiple PIPAC cycles, the abdomen is inspected laparoscopically prior to each repeat dose, and therefore the procedure has the added benefit of enabling the surgeon to monitor the patient’s progress and tailor continuing therapy accordingly. 

PIPAC is unique in that the technique is highly standardized with safety checklists, allowing for homogeneous practice across institutions and countries. Each prospective patient is discussed at a multidisciplinary tumor board prior to proceeding. Under general anesthesia, the patient is positioned supine, and two intraperitoneal ports are placed: a nebulizer port and an access port. The abdomen is insufflated with CO_2,_ and diagnostic laparoscopy is first performed to determine the initial PCI score, peritoneal biopsies are taken, and ascites fluid is drained and sent for cytology. A high-pressure injector is then connected to the nebulizer port. Pressurized aerosol chemotherapy drugs, typically cisplatin and doxorubicin, are administered through the port into the peritoneum and allowed to equilibrate for 30 min at 37 °C while maintaining capnoperitoneum. The aerosolized chemotherapy is then evacuated into a closed aerosol waste system, the catheter system removed, and port sites closed [70,71,72,73]. An example of a PIPAC set-up is shown in Figure 2.

### 4.2. Chemotherapy in PIPAC

Two chemotherapy regimens have been described for use in PIPAC: oxaliplatin monotherapy for colorectal cancer and cisplatin in combination with doxorubicin in other cancers with peritoneal metastases, including gastric cancer [74,75]. Specifically, gastric and ovarian tumor cells were found to be more sensitive to cisplatin and doxorubicin under PIPAC conditions in a preclinical study [76]. PIPAC cycles are administered at 6–8 week intervals, with a typical goal of reaching 3 cycles, and can be continued thereafter [70,77]. This can be given concurrently with systemic chemotherapy; a synergistic effect between the two has been proposed by Solass et al. [69], although some protocols recommend holding systemic treatment for two weeks perioperatively [68,76].

### 4.3. Safety and Outcomes

PIPAC has been studied in the context of peritoneal metastases, regardless of primary cancer, and has shown positive results overall [70,78]. In 2016, Nadiradze et al., who had opened a PIPAC program in Germany, were the first to study PIPAC in patients with gastric cancer and peritoneal carcinomatosis [79] (Table 2). There were few exclusion criteria, and most study patients had undergone prior palliative systemic chemotherapy (79%), were on 3rd or 4th line therapy (46%), and were not candidates for HIPEC or CRS due to poor functional status or advanced PCI. Of those, 50–60% had undergone prior gastrectomy [77,80]. Multiple studies have included concurrent systemic chemo with PIPAC. Although generally 3 cycles of PIPAC have been recommended, studies report that a significant proportion of patients are unable to undergo repeat dosing, primarily due to disease progression (though a small number did withdraw from the study or were lost to follow up) [77,81]. Although variable by study, between 24–71% of study patients were unable to receive at least 3 cycles of PIPAC [77,78,79,80,81,82]. In rare cases, the inability to access the PIPAC port due to adhesions have resulted in the termination of the procedure [79].

Patients who have undergone PIPAC typically have relatively advanced peritoneal disease, with median PCI scores of 14–19 at initial procedure, and are often symptomatic [77,78,79,80,81,82]. Some studies showed improvement of PCI with treatment [77,78]. Tumor response was noted in 40–60% of patients who underwent repeat procedures [77,79,81]. Most studies have described a median survival time of 6.7 to 15.4 months, although a study by Alyami in 2021 demonstrated median survival as high as 19 months [82]. One-year survival was reported around 50%. The PIPAC procedure itself has been well-tolerated, with usually a 3-day hospital stay and relatively low complication rates (3–15%). Importantly, patients with advanced peritoneal metastases maintained a good, stable quality of life when undergoing repeat PIPAC, with largely no gastrointestinal symptoms [75,83].

Interestingly, Girshally et al. conducted a retrospective study that suggested the possibility of neoadjuvant PIPAC ahead of CRS + HIPEC. A handful of patients with various primary tumors and extensive peritoneal disease who were not candidates for HIPEC (mean PCI 14.3) exhibited significant regression of disease after a mean of 3.5 cycles of PIPAC, and ultimately were able to undergo CRS with HIPEC [84]. This led the authors to posit that in select patients, 4 PIPAC cycles, which takes 5–6 months to complete, may transform diffuse peritoneal metastases to localized disease. This finding was also observed in the study by Alyami et al., in which 6 out of 42 patients with initially unresectable disease (median PCI 13) underwent 3 cycles of PIPAC with PCI downstaged to 3, and successfully underwent CRS + HIPEC [82].

Less than a decade in practice, PIPAC is still in its infancy, and a great deal is unknown about who should receive the treatment and its long-term effects. Data from randomized clinical trials are lacking, and currently, there are no published results from phase III trials. The use of PIPAC is recommended only at specialized centers in the context of clinical trials in Europe, as adoption in the United States and Asia has been limited thus far. Literature suggests it may be useful as a second- or third-line therapy for patients who have progressed through or are not candidates for certain treatments [81,82,85]. Overall, PIPAC has emerged as a promising alternative modality that is well tolerated and may offer a safe palliative or salvage option for patients who have failed other therapies. Additional clinical trial data is needed to assess its efficacy in the context of randomized studies.

## 5. Ongoing Trials

Intraperitoneal chemotherapy remains an exciting, developing field of research with a number of ongoing clinical trials (Table 3). Today, multiple phase III trials around the world are investigating the efficacy of HIPEC. GASTRIPEC (Germany, 2014–2021) enrolled patients with gastric or gastroesophageal junction cancer with peritoneal carcinomatosis and randomized them into two surgical groups: gastrectomy with peritonectomy only, or gastrectomy with peritonectomy and HIPEC, hypothesizing of survival advantage in the HIPEC cohort [86]. GASTRICHIP (France, 2013–2026) aims to compare 5-year survival rates of patients with advanced gastric cancer and positive peritoneal cytology randomized to either curative gastrectomy or gastrectomy with HIPEC [87]. PERISCOPE II (Netherlands, 2017–2022) is randomizing patients with gastric cancer and limited peritoneal disease into two study arms to undergo either gastrectomy with CRS and HIPEC or palliative systemic chemotherapy only and comparing survival outcomes [88]. Dragon II (China, 2019–2021) is the first randomized trial to investigate the neoadjuvant role of HIPEC: the trial compares the effect of a comprehensive multimodal intraperitoneal regimen (laparoscopic HIPEC, neoadjuvant systemic chemotherapy, CRS with HIPEC, adjuvant systemic chemotherapy) with standard therapy of gastrectomy with adjuvant systemic chemotherapy [89].

Clinical trials investigating PIPAC are also emerging. PIPAC EstoK 01 (France, 2019–2022) is a phase II trial that aims to compare survival and quality of life in patients who undergo PIPAC (with cisplatin and doxorubicin) and systemic chemotherapy versus systemic chemotherapy alone for gastric cancer with carcinomatosis [90]. PIPAC-OPC2 (Denmark, 2016–2020) is a similar trial investigating the outcomes of PIPAC on peritoneal carcinomatosis of both gastric and non-gastric cancer etiology [91]. Trials are also beginning to appear outside of Europe, including the first phase I PIPAC study in the United States, which is still in the accrual phase [92,93].

## 6. Conclusions

The application of intraperitoneal chemotherapy has evolved rapidly over the last three decades. Studies have shown encouraging results in improved survival thus far, suggesting the potential for multimodal therapy to reshape the management of gastric cancer with peritoneal metastases, which currently has a dismal prognosis. Studies also highlight the importance of preoperative workup and careful patient selection in trying to predict which patients will benefit from surgical resection and/or intraperitoneal therapies. With several currently ongoing clinical trials in HIPEC and PIPAC, the next few years will prove to be an exciting time as the results hopefully bring more clarity to the use and indications of this promising treatment.

## Figures and Tables

**Figure 1 cancers-14-00570-f001:**
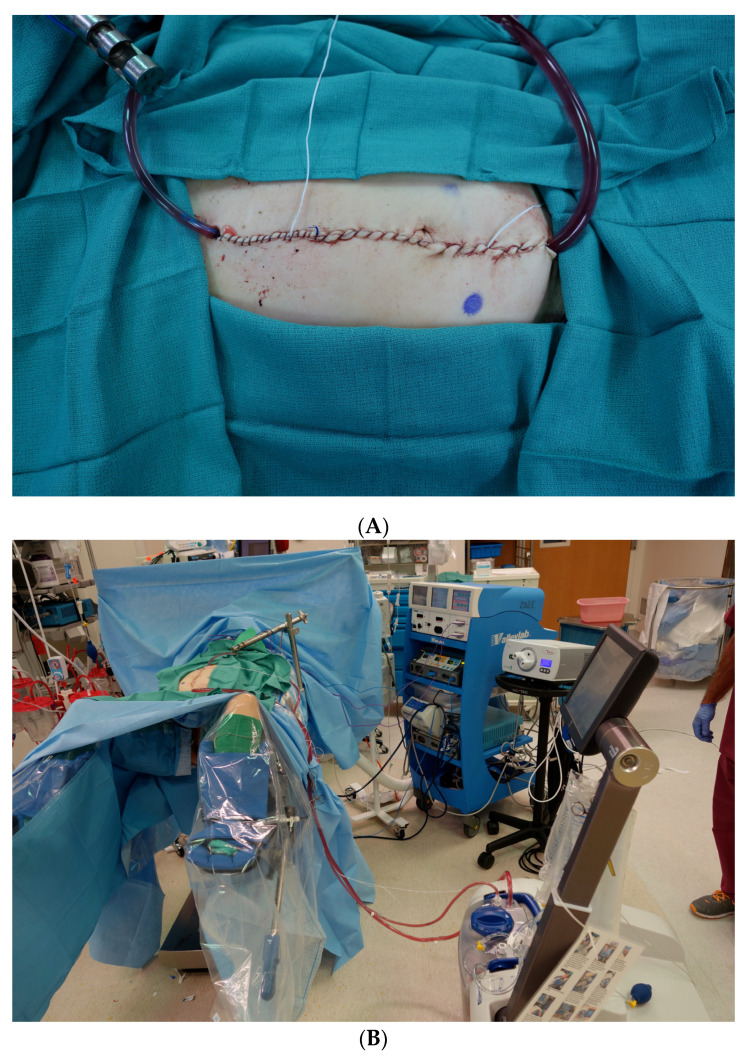
Hyperthermic Intraperitoneal Chemotherapy Perfusion During Cytoreductive Surgery. (**A**): Closed HIPEC Technique. The midline incision is temporarily closed to create a closed system for the infusion of chemotherapy into the peritoneal cavity. (**B**): HIPEC Room Set-Up. Operating room set-up during HIPEC, with inflow and outflow peritoneal catheters connected to the perfusion machine.

**Figure 2 cancers-14-00570-f002:**
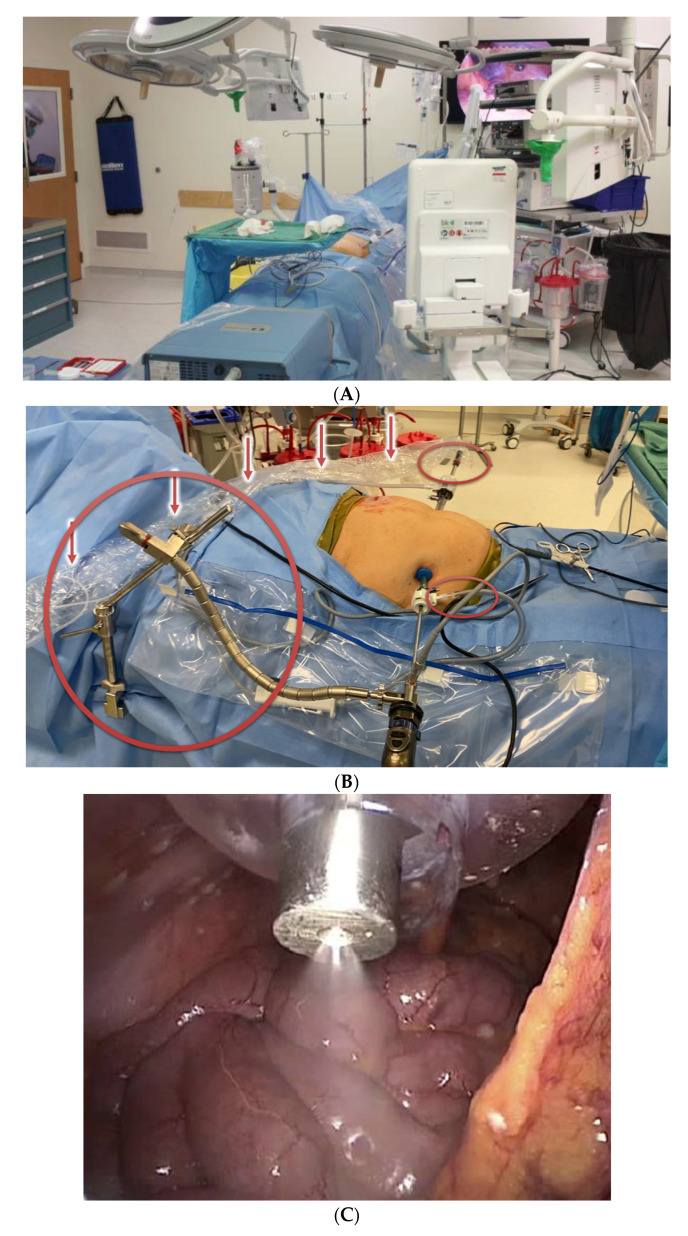
Intraoperative Administration of Pressurized Intraperitoneal Aerosolized Chemotherapy. (**A**): PIPAC Room Set-Up. The operating room is cleared during administration of PIPAC. Video and anesthesia monitors face adjacent rooms to ensure patient safety. (**B**): PIPAC Patient Set-Up. Self-retaining retractors are used to steady the laparoscope camera and Capnopen. Arrows indicate connection to power injector for delivery of chemotherapy. (**C**): Pressurized Aerosolized Chemotherapy Delivery. Laparoscopic visualization of the Capnopen during PIPAC.

**Table 1 cancers-14-00570-t001:** Summary of Studies on Hyperthermic and Normothermic Intraperitoneal Chemotherapy in Gastric Cancer with Peritoneal Metastases.

Study	Patients & Design	Chemotherapy Regimen	Complications	Survival Outcomes	Country
Yang et al. (2011) [24]	HIPEC: Randomized phase III trial (Randomized 1:1 to 2 arms)	Study arm: Open HIPEC with cisplatin (120 mg) and mitomycin C (30 mg), 60–90 min at 43 °C. Systemic chemo post-op Control arm: Systemic chemo post-op	Serious adverse events: CRS + HIPEC = 14.7%, CRS only = 11.7%	Disease-specific survival: CRS + HIPEC = 11.0 mo, CRS only = 6.5 mo (*p* <0.05)	China, single-center
	Gastric cancer with peritoneal metastases, median PCI 15				
	68 patients: 34 CRS + HIPEC, 34 CRS only				
GYMSSA Trial Rudloff et al. (2014) [29]	HIPEC: Randomized phase III trial (Randomized 1:1 to 2 arms)	Study arm: Closed HIPEC with oxaliplatin (460 mg/m^2^) with induction IV 5-FU (400 mg/m^2^) and leucovorin (20 mg/m^2^), for 30 min at 41 °C. Post-op FOLFOXIRI Control arm: Systemic chemo only (FOLFOXIRI)	Serious adverse events: CRS + HIPEC = 89% Mortality: 11%	Overall survival: CRS + HIPEC = 11.3 mo, systemic chemo only = 4.3 mo	United States, single-center
	Gastric cancer with peritoneal metastases				
	16 patients: 9 CRS + HIPEC, 7 systemic chemo only				
Glehen et al. (2010) [25]	HIPEC: Retrospective (1989–2007)	Study arm: HIPEC with various regimens (mitomycin C/cisplatin or oxaliplatin-based), or EPIC with mitomycin C/5-FU on post-op days 1–5	Serious adverse events: 27.8% (enteric fistula = 15.9%, re-operation = 14.0%) Mortality: 6.5%	Total overall survival: 9.2 mo, 43% 1-year survival Complete cytoreduction survival: 15 mo, 63% 1-year survival	France, multicenter
	Gastric cancer with peritoneal metastases				
	159 patients: single-arm, CRS + HIPEC and/or EPIC				
Rau et al. (2020) [26]	HIPEC: Retrospective (2011–2016)	Study arm: HIPEC with various regimens (mitomycin C vs. cisplatin vs. doxorubicin vs. oxaliplatin)	Serious adverse events: 17.0% Mortality: 5.1%	Overall survival: 13 mo Survival by PCI: 0–6 = 18 mo, 7–15 = 12 mo, >16 = 5 mo (*p* = 0.002)	Germany, multicenter
	Gastric cancer with peritoneal metastases, median PCI 8				
	235 patients: single-arm, CRS + HIPEC				
CYTO-CHIP Bonnot et al. (2019) [4]	HIPEC: Retrospective (1989–2014)	Study arm: HIPEC with various regimens per institution-specific protocol Control arm: No HIPEC	Serious adverse events: CRS + HIPEC = 53.7%, CRS only = 55.3%	Overall survival: CRS + HIPEC = 18.8 mo, CRS only = 12.1 mo (*p* = 0.005) 3-year recurrence free survival: CRS + HIPEC = 20.4%, CRS only = 5.9%	France, multicenter
	Gastric cancer with peritoneal metastases, median PCI 3				
	277 patients: 180 CRS + HIPEC,				
	97 CRS only				
PHOENIX-GC Trial Ishigami et al. (2018) [30]	NIPEC: Randomized phase III trial (Randomized 2:1 to study arm)	Study arm: Systemic chemo S-1 days 1–14, IV paclitaxel and IP paclitaxel (20 mg/m^2^) on days 1 and 8; 3-week cycles Control arm: Systemic chemo only: S-1 days 1–21, cisplatin on day 8; 5-week cycles	Serious adverse events: neutropenia: NIPEC = 50%, systemic chemo only = 30%	Overall survival: NIPEC = 17.7 mo, systemic chemo only = 15.2 mo (*p* = 0.08)	Japan, multicenter
	Gastric cancer with peritoneal metastases				
	164 patients: 114 NIPEC, 50 systemic chemo only				
Yamaguchi et al. (2013) [31]	NIPEC: Prospective phase II trial	Study arm: Systemic chemo S-1 days 1–14, IV paclitaxel and IP paclitaxel (20 mg/m^2^) on days 1 and 8; 3-week cycles	Serious adverse events: neutropenia 34%	Overall survival: 17.6 mo 1-year survival: 77.1%	Japan, single-center
	Gastric cancer with peritoneal metastases				
	35 patients: single-arm, NIPEC				
Shinkai et al. (2018) [32]	NIPEC: Prospective phase II trial	Study arm: IP paclitaxel (60 mg/m^2^), followed by systemic chemo S-1 days 1–14, IV paclitaxel/IV cisplatin on days 1 and 8; 3-week cycles	Serious adverse events: none	Overall survival: 23.9 mo 1-year survival: 82.4% Regression of peritoneal metastases: 73.3%	Japan, single-center
	Gastric cancer with peritoneal metastases				
	17 patients: single-arm, NIPEC				
Cheong et al. (2007) [33]	NIPEC: Prospective cohort study	Study arm: EPIC with 5-FU (500 mg/m^2^) and cisplatin (40 mg/m^2^) for 60 min, 4 consecutive days; every 4 weeks for 12 cycles	Serious adverse events: 22.7% Mortality: 2.6%	Overall survival: 11.4 mo Survival by residual tumor: R0 = 25.5 mo, R1 = 15.6 mo, R2 = 7.2 mo (*p* < 0.001)	Korea, single-center
	Advanced gastric cancer with or without peritoneal metastases				
	154 patients: gastrectomy with D2 lymphadenectomy + NIPEC				
INPACT Trial Takahashi et al. (2018) [34]	NIPEC: Randomized phase II trial (Randomized 1:1 to 2 arms)	Study arm: IP paclitaxel (60 mg/m^2^) on weekly basis for 7 doses, then systemic chemo (S-1 or S-1/cisplatin) Control arm: IV paclitaxel on weekly basis for 7 doses, then systemic chemo	Serious adverse events: low, similar in both groups	Overall survival: NIPEC = 42.3 mo, systemic chemo = 37.7 mo (*p* = 0.63) 2-year progression-free survival: NIPEC = 38.4%, systemic chemo = 46.6% (*p* = 0.53)	Japan, multicenter
	Gastric cancer with minimal peritoneal metastases or positive cytology				
	83 patients: 39 gastrectomy + NIPEC, 44 gastrectomy + systemic chemo				

Select studies evaluating the efficacy of intraperitoneal chemotherapy modalities, including HIPEC and NIPEC, in the treatment and palliation of gastric cancer with peritoneal metastases. CRS = cytoreductive surgery, HIPEC = hyperthermic intraperitoneal chemotherapy, NIPEC = normothermic intraperitoneal chemotherapy, EPIC = early postoperative intraperitoneal chemotherapy, IP = intraperitoneal, PCI = peritoneal carcinomatosis index, FOLFOXIRI = 5-FU, leucovorin, oxaliplatin, irinotecan).

**Table 2 cancers-14-00570-t002:** Summary of Studies on Pressurized Intraperitoneal Aerosolized Chemotherapy in Gastric Cancer with Peritoneal Metastases.

Study	Patients & Design	Chemotherapy Regimen	Complications	Survival Outcomes	Country
Nadiradze et al. (2016) [79]	PIPAC: Retrospective (2011–2013)	Study arm: Cisplatin (7.5 mg/m^2^) and doxorubicin (1.5 mg/m^2^) at 12 mmHg for 30 min at 37 °C, may repeat	Serious adverse events: 12.5% Mortality: 8.3%	Overall survival: 15.4 mo Histological tumor response: 50%	Germany, single-center
	Gastric cancer with peritoneal metastases, therapy-resistant				
	24 patients: single-arm, PIPAC				
Alyami et al. (2017) [78]	PIPAC: Retrospective (2015–2016)	Study arm: Doxorubicin (1.5 mg/m^2^) and cisplatin (7.5 mg/m^2^) for non-colorectal cancer; oxaliplatin (92 mg/m^2^) or mitomycin C (1.5/m^2^) for colorectal, for 30 min at 12 mmHg; repeat every 6–8 weeks for at least 3 cycles	Serious adverse events: 9.7% Mortality: 6.8%	Decreased PCI: 64.5%	France, multicenter
	Non-resectable peritoneal metastases, various GI primary, median PCI 19				
	73 patients: single-arm, PIPAC				
PIPAC-GA2 Khomyakov et al. (2018) [81]	PIPAC: Prospective phase II trial	Study arm: 4 cycles of systemic chemo (XELOX); then PIPAC with doxorubicin (1.5 mg/m^2^) and cisplatin (7.5 mg/m^2^) at 12 mmHg for 30 min at 37 °C; every 6 weeks with 2 cycles XELOX between PIPAC cycles	Serious adverse events: none	Overall survival: 13 mo Major pathologic response: 60%	Russia, single-center
	Gastric cancer with peritoneal metastases, mean PCI 16				
	31 patients: single-arm, PIPAC				
Struller et al. (2019) [77]	PIPAC: prospective phase II trial	Study arm: Doxorubicin (1.5 mg/m^2^) and cisplatin (7.5 mg/m^2^) at 12 mmHg for 30 min at 37 °C; repeat every 6 weeks for 3 cycles	Serious adverse events: none	Overall survival: 6.7 mo Stable or disease regression: 40%	Germany, single-center
	Gastric cancer with peritoneal metastases, therapy-resistant now on salvage therapy, mean PCI 15				
	25 patients: single-arm, PIPAC				
Alyami et al. (2021) [82]	PIPAC: Retrospective (until 2018)	Study arm: Doxorubicin (1.5 mg/m^2^) and cisplatin (7.5 mg/m^2^) at 12 mmHg for 30 min at 37 °C; repeat every 6–8 weeks for at least 3 cycles	Serious adverse events: 6.1% Mortality: 4.7%	Overall survival: 19.1 mo 14.3% resectable after PIPAC treatment	France, single-center
	Gastric cancer with unresectable peritoneal metastases, median PCI				
	42 patients: single-arm, PIPAC				

Select studies evaluating the efficacy of PIPAC in the treatment and palliation of gastric cancer with peritoneal metastases. PIPAC = pressurized intraperitoneal aerosolized chemotherapy, PCI = peritoneal carcinomatosis index, XELOX = capecitabine, oxaliplatin.

**Table 3 cancers-14-00570-t003:** Summary of Ongoing Clinical Trials in Intraperitoneal Chemotherapy in Gastric Cancer.

Clinical Trial	Design	Estimated Inclusion & Enrollment	Surgical Resection	Chemotherapy Regimen	Survival Metrics	Country
**GASTRIPEC** Rau et al. NCT02158988	HIPEC: Randomized phase III trial	Gastric cancer with peritoneal metastases 105 total, randomized 1:1 to 2 arms	Study arm: Gastrectomy with CRS + HIPEC Control arm: Gastrectomy with CRS	Study arm: 3 cycles systemic chemo, followed by CRS + HIPEC with mitomycin C (15 mg/and cisplatin (75 mg/m^2^) for 60 min at 41–43 °C, then 3 cycles systemic chemo post-op Control arm: 4 cycles systemic chemo, followed by CRS only, then 3 cycles systemic chemo post-op	Primary endpoint: 2-year overall survival Secondary endpoints: 30-day morbidity, time to disease progression, quality of life	Germany, multicenter
**GASTRICHIP** Glehen et al. NCT01882933	HIPEC: Randomized phase III trial	T3-T4 gastric cancer with positive nodes and/or cytology 306 total, randomized 1:1 to 2 arms	Study arm: Gastrectomy, D1-D2 lymphadenectomy + HIPEC Control arm: Gastrectomy, D1-D2 lymphadenectomy	Study arm: Folinic acid and 5-FU (400 mg/m^2^) IV induction chemo 15 min prior to HIPEC with oxaliplatin (250 mg/m^2^) for 30 min at 42–43 °C. Control arm: No HIPEC	Primary endpoint: 5-year overall survival Secondary endpoints: 3 and 5-year recurrence-free survival, recurrence site, morbidity, quality of life	France, multicenter
**PERISCOPE II** Sandick et al. NCT03348150	HIPEC: Randomized phase III trial	T3-T4 gastric cancer with limited peritoneal metastases, PCI < 7 182 total, randomized 1:1 to 2 arms	Study arm: Gastrectomy with CRS + HIPEC Control arm: None	Study arm: 3–4 cycles of systemic chemo (variable regimens), followed by CRS + HIPEC with oxaliplatin (460 mg/m^2^) for 30 min at 41–42 °C, then docetaxel (50 mg/m^2^) for 90 min at 37 °C Control arm: Systemic chemo (variable regimens)	Primary endpoint: 5-year overall survival Secondary endpoints: 5-year progression-free survival, treatment toxicity, cost, quality of life	Netherlands, multicenter
**Dragon II** Zhu et al. ChiCTR1900024552	HIPEC: Randomized phase III trial	T4 gastric cancer, with no peritoneal metastases or cytology 326 total, randomized 1:1 to 2 arms	Study arm: NLHIPEC, gastrectomy with D2 lymphadenectomy + HIPEC Control arm: Gastrectomy with D2 lymphadenectomy	Study arm: NLHIPEC with paclitaxel (80 mg/m^2^) for 60 min at 43 °C, followed by 3 cycles IV chemo (SOX), gastrectomy + HIPEC with paclitaxel (80 mg/m^2^), then 5 cycles IV chemo (SOX) post-op Control arm: Gastrectomy followed by 8 cycles of IV chemo (SOX)	Primary endpoint: 5-year progression-free survival Secondary endpoints: 5-year overall survival, peritoneal metastasis rate, morbidity	China, multicenter
**PIPAC EstoK 01** Eveno et al. NCT04065139	PIPAC: Randomized phase II trial	Gastric cancer with peritoneal metastases, PCI > 8 94 total, randomized 1:1 to 2 arms	Study arm: None Control arm: None	Study arm: Doxorubicin (2.1 mg/m^2^) and cisplatin (10.5 mg/m^2^) for 30 min at 12 mmHg and 37 °C, followed by 2 cycles of IV chemo (variable regimens). Total 3 PIPAC cycles, 6–8 weeks apart Control arm: Systemic chemo (variable)	Primary endpoint: 2-year progression-free survival Secondary endpoints: 2-year overall survival, morbidity, quality of life, resectability rate	France, multicenter
**PIPAC-OPC2** Mortensen et al. NCT03287375	PIPAC: Randomized phase II trial	Peritoneal metastases from GI or ovarian primary 137 total, single-arm	Study arm: None	Study arm: Doxorubicin (1.5 mg/m^2^), cisplatin (7.5 mg/m^2^) non-colorectal cancer or oxaliplatin (92 mg/m^2^) for colorectal, for 30 min at 12 mmHg. Systemic chemo variable. Total 3 PIPAC cycles	Primary endpoint: 4-year major disease response Secondary endpoints: Quality of life, utility of MRI to evaluate treatment response	Denmark, single-center

A sample of several ongoing registered clinical trials across several countries examining the efficacy of various intraperitoneal chemotherapy regimens in the treatment of advanced and metastatic gastric cancer. CRS = cytoreductive surgery, HIPEC = hyperthermic intraperitoneal chemotherapy, PIPAC = pressurized intraperitoneal aerosolized chemotherapy, PCI = peritoneal carcinomatosis index, NLHIPEC = neoadjuvant laparoscopic hyperthermic intraperitoneal chemotherapy, SOX = combination S-1 and oxaliplatin chemotherapy.

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
