# Peer review of "Review of Regional Therapies for Gastric Cancer with Peritoneal Metastases"

_cancers, 2022, doi:10.3390/cancers14030570_

Round 1

Reviewer 1 Report

This is a well written review that nicely describes the landscape of treatments for gastric cancer with peritoneal metastases 

This is an interesting topic and there is considerable controversy about the optimal management of these patients thus this review is important 

I do not have any major comments with regards to the manuscript 

Given the fact that many physicians might not be familiar with the set up of HIPEC, NIPEC and PIPAC consider adding a picture/schema illustrating the setup so the non expert reader can understand the technical differences 

Also i recommend adding a brief comment on the status of PIPAC in the United States (trials in process etc) 

Author Response

Dear Reviewer,

Thank you for the constructive comments. Please see our responses below:

We have included photos to illustrate both HIPEC and PIPAC setups, in hopes that this will help clarify the process to those are not familiar with intraperitoneal therapies.

To our knowledge, there is only one phase I clinical trial on PIPAC in the United States at the present time, which is still in the accrual phase. We discuss this in lines 513-515.  

Thank you again for considering our manuscript.

Reviewer 2 Report

This is a very fine review of treatment strategies for gastric cancer with peritoneal metastases. It is a well-balanced, comprehensive overview of trials applying local therapeutic regimen in carcinosis peritonei from gastric cancer.

Author Response

Dear Reviewer,

Thank you for the comments and for reviewing our manuscript.

Reviewer 3 Report

In this manuscript, Dr Sun and Dr Lee try to summarize different intraperitoneal chemotherapy treatment regimens for gastric cancer with peritoneal metastases . The text is simplistic and not intriguing, English needs major revisions. Furthermore, the correct indication of the abbreviations used is missing.

Manuscript is lacking of correct description of prognosis of peritoneal involvement of gastric cancer, and  there is no reference to the theme of oligometastatic disease, with the implication of intraperitoneal chemo.

No mention about differences in responses, survivals and implications between Asian and Western patients is reported.

Manuscript is also lacking of a conclusive discussion and appears like a list of studies no properly discussed and without author’s point of view.

In this form the manuscript  does not add a systematic discussion or an important contribution for scientific community

Author Response

Dear Reviewer,

Thank you for consideration of our paper. We have edited the manuscript with an intent on improving the content and language. Abbreviations and definitions have been checked and corrected. As this was an invited review from the editor, Dr. Jeffrey Norton, the task was to summarize regional therapies for peritoneal metastasis from gastric cancer. We did not include data on oligometastatic disease to other sites or differences in Asian vs. Western populations. While this is no doubt important in choosing therapies for this disease, there is no specific link to differences in outcomes when peritoneal carcinomatosis is present.

Regarding the paper’s value to the scientific community, we will once again state that this is an invited review. Our intent was not to publish this as an original piece, but to provide a summary of clinical trials utilizing these emerging therapies. There is no solid data, and we have addressed this several times. Much like HIPEC for colorectal cancer, ovarian cancer, and mesothelioma, early data may lead to future randomized studies, which is what this field desperately needs.

Thank you again for reviewing and reconsidering our manuscript.

Reviewer 4 Report

We congratulate Beatrice Sun and Byrne Lee to this narrative review about hyperthermic intraperitoneal chemotherapy (HIPEC), normothermic intraperitoneal chemotherapy (NIPEC) and pressurized intraperitoneal chemotherapy (PIPAC) as regional treatment options for gastric cancer peritoneal metastasis.

The authors provide a broad overview about the historic development of regional chemotherapeutic delivery to the peritoneum and current trends and issues on this topic. They analyze some of the most cited studies concerning CRS and HIPEC, although they discussed the older studies such as the GYMSSA trial from 2011 more extensively than newer studies, which would for sure increase the benefit for the readers of this review. While this review is well-written and comprehensive, there are several narrative reviews on this same topic even within the last year, and the necessity for another review without further published trials remains vague.

Major points:

  1. There is a table giving a “summary of major studies on intraperitoneal chemotherapy in gastric cancer with peritoneal metastases”, but there is a no definition of a “major study” provided. I would suggest to separate randomized controlled trials from retrospective trials more visibly in the table, or divide it into two separate tables.
  2. It should be emphasized more, that there is little proof (from reliable, randomized controlled trials) for the efficacy of intraperitoneal chemotherapy in gastric cancer. While the authors do mention the controversy several times, the reasons for this controversy should be discussed more specifically, maybe in a separate section - such as the fact that all non-randomized controlled trials displayed here are immensely biased and not fit to show any survival benefit. 
  3. The sections on survival and complications (in 2.1.4) should be separated. It is redundant, as the survival / efficacy has already been discussed (appropriately) in the chapters on randomized and non-randomized trials before. I would also recommend to separate data on efficacy and safety for the other treatment methods. Safety and complications are studied quite well for intraperitoneal chemotherapies, while data survival / efficacy are scarce and should be viewed with caution and discussed openly.

Minor points

  1. Some concepts tend to be repeated in every review on this topic, like the “tumor cell entrapment” that Sugarbaker described (in 1996, not 2003, line 54), and that, to my knowledge, has not been confirmed in more recent, meaningful, experimental studies. Another one is the “poor penetration of systemic chemotherapy in peritoneal disease” line 60-62. It would be refreshing and interesting to actually present proof for this hypothesis.
  2. Concerning safety and complications after CRS and HIPEC, the authors could present more recent data on strategies to prevent complications such as sodium thiosulfate for nephroprotection in Cisplatin-based HIPEC regimen, or studies about the interval between last dose of systemic chemotherapy and PIPAC or the use on antibodies in combination with PIPAC.

Author Response

Dear Reviewer,

Thank you for the thoughtful comments and critiques. Please see our responses below.

Major points:

  1. Thank you for the comment. As there are very few randomized controlled trials in HIPEC, NIPEC and PIPAC, we have elected to separate the original Table 1 into 2 separate tables by modality for ease of reading: Table 1 now includes studies on HIPEC and NIPEC, and Table 2 now includes studies on PIPAC. It is difficult to define a “major” study in this field where there is limited literature and we have thus removed “major” from the table title.
  2. The reviewer raises a good point that there is little proof of survival benefit of HIPEC in gastric cancer with peritoneal metastases. Unfortunately, there is only one completed trial in that studies HIPEC in gastric cancer with peritoneal metastases, so the ability to draw conclusions such as survival benefit from clinical trials is limited. Although a number of non-randomized studies, due to their retrospective nature and patient selection criteria subject them to bias, their findings suggest a potential for survival benefit. While this is not proof of increased survival, the trend towards improved survival is promising and we believe it is worth discussing. In our review, we do raise caution on several occasions regarding the interpretation of results of non-randomized HIPEC studies, including lines 182-188 and 199-204. We agree that this highlights the need for ongoing/future clinical trials to address this gap and inform survival benefit in the future.
  3. Thank you for the organizational suggestion. We have separated out sections on HIPEC complications and HIPEC survival (lines 198 and 219, respectively). The safety of intraperitoneal therapies is important, and we have ensured that these are addressed: NIPEC (lines 295-296, 310-311), and PIPAC (section 4.3). Again, the authors have made an effort to emphasize that survival trends are promising but should be interpreted with caution given the inherent biases from retrospective studies included in this review.

Minor points:

  1. We have elected to remove this section, as this review paper is aimed at summarizing regional therapies for peritoneal metastasis from gastric cancer, and the basic science mechanism behind tumor cell penetration is not the focus of our paper.
  2. Thank you for this suggestion. We have added a brief discussion on the nephroprotection with sodium thiosulfate (lines 212-216). The interval between systemic chemotherapy and PIPAC is discussed in lines 421-423. To our knowledge, we are not aware of any studies or trials regarding antibodies and intraperitoneal therapies in gastric cancer.

Thank you again for reviewing and considering our manuscript.  

Round 2

Reviewer 3 Report

Dr.Lee and Dr. Sun edited the manuscript with a substantial improvement of  contents . The aim of the paper is to summarize regional therapies for peritoneal metastasis from gastric cancer, without data about oligometastatic forms. The work is now a good  summary of clinical trials and  the improvements done make it suitable for pubblication.